# 16S rRNA gene sequencing of rectal swab in patients affected by COVID-19

Antonio Mazzarelli[1], Maria Letizia Giancola[1], Anna Farina[1], Luisa Marchioni[1], Martina Rueca[1], Cesare Ernesto Maria Gruber[1], Barbara Bartolini[1], Tommaso Ascoli Bartoli[1], Gaetano Maffongelli[1], Maria Rosaria Capobianchi[1], Giuseppe Ippolito[1], Antonino Di Caro[1] *, Emanuele Nicastri[1‡], Valerio Pazienza[2‡], INMI COVID-19 study group[¶]

**1** National Institute for Infectious Diseases, INMI "Lazzaro Spallanzani", IRCCS, Rome, Italy, **2** Division of Gastroenterology, Fondazione-IRCCS "Casa Sollievo della Sofferenza" Hospital, S. Giovanni Rotondo (FG), Italy

☯ These authors contributed equally to this work.
‡ These authors also contributed equally to this work.
¶ Membership of the Collaborators Members of the National Institute for Infectious Diseases (INMI) COVID-19 study group is provided in the acknowledgments.
* antonino.dicaro@inmi.it

**Data Availability Statement:** Raw 16S sequencing data are available in the NCBI Sequence Read Archive under Accession Number PRJNA681516 (https://www.ncbi.nlm.nih.gov/bioproject/PRJNA681516).

## Abstract

COronaVIrus Disease-2019 (COVID-19) is a pandemic respiratory infection caused by a new betacoronavirus, the Severe Acute Respiratory Syndrome-CoronaVirus-2 (SARS-CoV-2). Few data are reported on the gut microbiota in COVID-19 patients. 16S rRNA gene sequencing was performed to reveal an altered composition of the gut microbiota in patients with COVID-19 pneumonia admitted in intensive care unit (ICU) (i-COVID19), or in infectious disease wards (w-COVID19) as compared to controls (CTRL). i-COVID19 patients showed a decrease of Chao1 index as compared to CTRL and w-COVID19 patients indicating that patients in ICU displayed a lower microbial richness while no change was observed as for Shannon Index. At the phylum level, an increase of Proteobacteria was detected in w-COVID19 patients as compared to CTRL. A decrease of *Fusobacteria* and *Spirochetes* has been found, with the latter decreased in i-COVID19 patients as compared to CTRL. Significant changes in gut microbial communities in patients with COVID-19 pneumonia with different disease severity compared to CTRL have been identified. Our preliminary data may provide valuable information and promising biomarkers for the diagnosis of the disease and, when validated in larger cohort, it could facilitate the stratification of patients based on the microbial signature.

## 1. Introduction

COronaVIrus Disease-2019 (COVID-19) is a pandemic respiratory infection caused by a new betacoronavirus infection, the Severe Acute Respiratory Syndrome-CoronaVirus-2 (SARS-CoV-2). It may progress rapidly to acute respiratory distress syndrome with remarkable morbidity and mortality [1]. However, SARS-CoV-2 can be detected in specimens from different

**Funding:** This study was supported by funds to the Istituto Nazionale per le Malattie Infettive (INMI) Lazzaro Spallanzani IRCCS, Rome, Italy, from the Ministero della Salute (Ricerca Corrente, linea 1; COVID- 2020-12371817), the European Commission – Horizon 2020 (EU project 101003544 – CoNVat; EU project 101003551 – EXSCALATE4CoV; EU project 101005111 DECISION; EU project 101005075-Unità Operativa Complessa Microbiologia e Banca Biologica Direttore: Dr. Antonino Di Caro- Tel. 0655170685 Fax 065594555 KRONO) and the European Virus Archive – GLOBAL (grants no. 653316 and no. 871029). Valerio Pazienza is supported by Italian Association for Cancer Research (AIRC) under IG 2019 - ID. 23006 project – P.I. The funders had no role in study design, data collection and analysis, decision to publish, or preparation of the manuscript.

**Competing interests:** The authors have declared that no competing interests exist.

sites and therefore it could potentially be transmitted in other ways than respiratory droplets [2]. In this regard, recent studies reported that SARS-CoV-2 RNA has been found in anal swabs, meaning that the virus could potentially be transmitted also through oral-fecal route [3]. These findings might suggest that other organs apart from lung might be additional sites for virus entry, repository and/or replication. Gastrointestinal (GI) symptoms, such as diarrhea (2%-10.1%), nausea and vomiting (1%-3.6%), are not very common at present in COVID-19 patients [4]. Nevertheless, an important proportion of patients were observed during the global pandemic showing atypical gastrointestinal symptoms [5].

Recently, several studies have demonstrated that respiratory infections are associated with changes of the gut microbiota composition [6, 7]. Typically, *Bacteroidetes* and *Firmicutes* prevail in the gut microbiota while potentially pathogenic species, such as some of those belonging to the phylum *Proteobacteria*, are present in a minor percentage [8, 9]. Current studies evaluate the relationship between the lung and the GI microbiota but this connection is not completely understood [10]. Patients with respiratory infections generally have gut dysfunction, which is related to a more severe clinical course of the disease. This phenomenon can also be observed in COVID-19 patients [5].

The gut microbiota has been shown to affect pulmonary health through cross-talk between the gut microbiota and the lungs, which is referred to as the "gut-lung axis" [11]. However, the "gut-lung axis" is supposed to be bidirectional: the gut microbiota, through microbial products and immune-modulators released upon recognition of commensals and pathogens by intestinal immune cells, can regulate lung immunity, influence the lung microbiota, and vice versa [12, 13]. Previous studies have shown that the modulation of the gut microbiota can reduce the severity of enteritis and ventilator-associated pneumonia by interacting with early replication of the viruses in the pulmonary epithelium, as in the case of influenza virus [14].

Angiotensin-converting enzyme 2 (ACE2) is the main receptor of SARS-CoV [14] and of SARS-CoV-2 [15]. This receptor is highly expressed in both the respiratory tract and GI, so it is possible to consider that SARS-CoV-2 uses ACE2 receptor to get into both body districts [16, 17].

All these virus characteristics raise a remarkable possibility that the pulmonary disease caused by SARS-CoV-2 may influence the gut microbiota [5].

In this pilot study, we performed the 16S RNA sequencing of fecal samples from COVID-19 in patients admitted to the National Institute for Infectious Disease (INMI) L. Spallanzani in Rome, Italy, between April 15, 2020 and May 31, 2020.

## 2. Methods

### 2.1. Study design

All the subjects agreed to participate according to the ethical guidelines of the 2013 Declaration of Helsinki signing an informed consent under the Ethical committee of the National Institute for Infectious Diseases Lazzaro Spallanzani—IRCCS approval number (n. 9/2020; n. 3 23.12.2019) and followed the same pre-analytical and analytical procedures, including fecal samples collection and storage. Data were analyzed anonymously.

From April 15 2020 and May 31 2020, rectal swabs were collected from patients hospitalized to INMI Spallanzani in Rome with confirmed or suspected SARS-CoV-2 infections. Fifteen out of 23 patients were affected by COVID-19 while 8 patients were COVID-19 negative.

All patients had pneumonia and were classified in three groups: a) COVID-19 patients with nose-pharyngeal swab positive for SARS-CoV-2, >18 years of age admitted in infectious disease wards at the time of the rectal swab execution (w-COVID19); b) COVID-19 patients with nose-pharyngeal swab positive for SARS-CoV-2, >18 years of age admitted in ICU (i-

COVID19); c) 'controls': patients admitted in the same time period with nose-pharyngeal swab negative for SARS-CoV-2 infections, hospitalized in ICU and/or in infectious disease ward (CTRL). We performed the rectal swab one or two days after the hospitalization.

Patient data, including laboratory test results and clinical manifestations, were obtained from Laboratory Information Systems (LIS) and clinical records.

Rectal swabs from COVID-19 patients were processed in the laboratory within 4 hours after collection or stored at -80°C until analysis.

## 2.2. DNA extraction

Samples were treated with 500 μl of Lysis Buffer ATL (QIAGEN, Hilden, Germany) at 56°C for 10 min with 20 μl Proteinase K (Darmstadt, Germany) before DNA extraction. Microbial DNA was extracted from 500 μl of sample using Qiasymphony automatic extractor (QIAGEN, Hilden, Germany) according to the manufacturer's protocol.

## 2.3. S sequencing and analysis

DNA samples were generated from PCR amplicons targeting the hypervariable regions V2, V4, V8 and V3-6, 7–9 of the 16S gene and libraries were processed using the Ion 16S metagenomics Kit. Ion Xpress Plus Fragment Library kit was used for libraries obtainment. Sequencing was performed on Ion 530 chip by Ion S5 sequencer (Ion Torrent-ThermoFisher Scientific).

The sequencing run has generated in total $16 \times 10^6$ reads with the 77% of high quality reads (21% low quality, 2% test fragments). Finally we obtained $5.5 \times 10^5$ reads per sample (reads length were 244 bp mean, 260 bp median and 289 bp mode) and all analyzed specimens showed a suitable library's profile. The analysis was performed by 16S Metagenomics GAIA 2.0 software and DESeq2 package software. Sequence data generated as FASTQ files, were analyzed using the 16S Metagenomics GAIA 2.0 software which performs the quality control of the reads/pairs (i.e., trimming, clipping and adapter removal steps) through FastQC and BBDuk. The reads/pairs are mapped with BWA-MEM against the 16S databases (GAIA based on NCBI). Differential expression analysis using DESeq2 package to test for differential analysis by use of negative binomial generalized linear models was used. Only changes with FDR below 0.05 were considered significant. The percent similarity used to determine species and genus calls was 93% at genus, 97% at species. PCoA analysis was obtained with GAIA software based on Bray-Curtis dissimilarities.

## 2.4. Statistical analysis

To evaluate if any clinical or laboratory variables are significantly different between patient cohorts, one-way ANOVA test and Kruskal-Wallis rank sum test were performed in R environment (www.cran.r-project.org) using aov and kruskal.test functions, respectively. Venn diagrams were obtained with Venny 2.1.0.

# 3. Results

## 3.1. Study population

The study population included 23 hospitalized inpatients; 15 out of 23 were patients with confirmed SARS-CoV-2 infection, 9 w-COVID19 and 6 i-COVID19, 8 were CTRL (three hospitalized in ICU and five in floor). Clinical data of the study patients are shown in Table 1. Overall, thirteen patients (56%) were male; median age was 67 (IQR 44–83). All patients presented pneumonia (for our CTRL six out of eight had bacteria pneumonia while 2 CTRL had non-COVID-19 viral pneumonia) and none of them had diarrhea when the rectal swab was performed (one or two days after the hospitalization). ANOVA and Kruskal-Wallis analysis in

**Table 1. Clinical and serologic data of the 23 enrolled patients.**

| Variables | w-COVID19 | i-COVID19 | CTRL | ANOVA p-values[*] | Kruskal-Wallis p-values |
|---|---|---|---|---|---|
| Number of patients (N = 23) | 9 | 6 | 8 | - | - |
| Median age, years (IQR) | 67 (IQR 44–83) | 70 (64–74) | 69 (51–77) | 0.702 | 0.7165 |
| Male, n (%) | 5 (55%) | 3 (50%) | 5 (62%) | - | - |
| Comorbidities, n (%) | 7 (78%) | 6 (100%) | 6 (75%) | - | - |
| Coinfections, n (%) | 1 (11%) | 4 (67%) | 0 (0%) | - | - |
| Colonization, n (%)[‡] | 2 (22%) | 3 (50%) | 0 (0%) | - | - |
| | *E. Faecium, E. Faecalis* | *2 E. Faecium, 1 E. Faecalis* | | | |
| Antibiotic therapy at swab, n (%) | 5 (55%) | 3 (50%) | 3 (37%) | - | - |
| Lymphocytes, mm³ (IQR) | 1310 (1190–1480) | 810 (437–995) | 1400 (950–2070) | 0.201 | 0.1503 |
| C-reactive protein, mg/dl (IQR) | 3.25 (0.84–6.54) | 10.94 (3.09–12.53) | 1.45 (0.92–5.04) | 0.228 | 0.2711 |
| Ferritin, ng/ml (IQR) | 393 (169–616) | 960 (565–1121) | 289 (88–251) | 0.0153 | 0.04481 |
| Fibrinogen, mg/dl (IQR) | 529 (422–616) | 469 (346–600) | 470 (332–527) | 0.751 | 0.6684 |
| D-dimer, ng/ml (IQR) | 715 (265–1760) | 638 (350–865) | 760 (442–1845) | 0.309 | 0.7042 |

Table 1: w-COVID19: patients affected by COVID-19 hospitalized in ward for highly infectious diseases; i-COVID19: patients hospitalized in intensive care unit; CTRL: patients admitted in the same time period with nose-pharyngeal swab negative for SARS-CoV-2 infections; IQR: interquartile range.

[*]p-value, P <0.05 are considered statistically significant. ± Enterococcus faecium and Enterococcus faecalis.

Table 1, showed that the p-values of the variables between our patient cohorts was not significant except for ferritin level that is significantly lower in ICU patients.

Eleven patients (48%) were receiving antibiotic therapy one or at most two days before the rectal swab was collected: 5 w-COVID19, 3 i-COVID19 and 3 CTRL.

Nineteen patients (83%) had one or more comorbidities, mainly cardiovascular and brain disorders. A concomitant infection (5 patients) and/or colonization (5 patients, although harbor a potentially pathological bacterium in their intestines namely *Enterococcus faecium* and *Enterococcus faecalis*, without developing the disease and consequently without the need of therapies or further tests) was found in seven patients (30%), 5 of them were hospitalized in ICU.

## 3.2. Microbial richness and diversity indices in COVID-19 patients as compared to CTRL

To understand the gut microbiota alterations between w-COVID19 and i-COVID19 patients with CTRL, as a first step we evaluated the richness and Shannon indices among the different groups. As shown in Fig 1A, Chao1 index was significantly decreased in i-COVID19 as

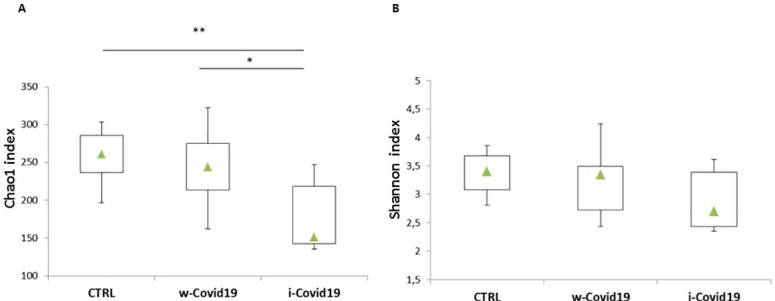

**Fig 1.** Box-plots of Chao1 index of species richness (A) and Shannon index of species diversity (B) in w-COVID19, i-COVID19 patients and CTRL. Triangles indicate the medians and Q1 and Q3 are reported.

compared to w-COVID19 ($p$ = 0.02) and CTRL ($p$ = 0.006). The same trend was also observed for Shannon index without reaching the statistical significance (Fig 1B).

Principal coordinates analysis (PCoA) was performed to cluster the microbial communities at the Family operational taxonomic unit (OTU) level based on Bray-Curtis distances. Fig 2 displayed distinct patterns among the three groups CTRL (red) w-COVID19 (green) and i-COVID19 (blue).

### 3.3. Microbiota profiles of w-COVID19 and i-COVID19 patients as compared to CTRL

At the Phylum level, *Proteobacteria* were significantly increased in w-COVID19 patients as compared to CTRL (17.1% vs 11.3% respectively FDR = 0.03) while *Spirochaetes* and

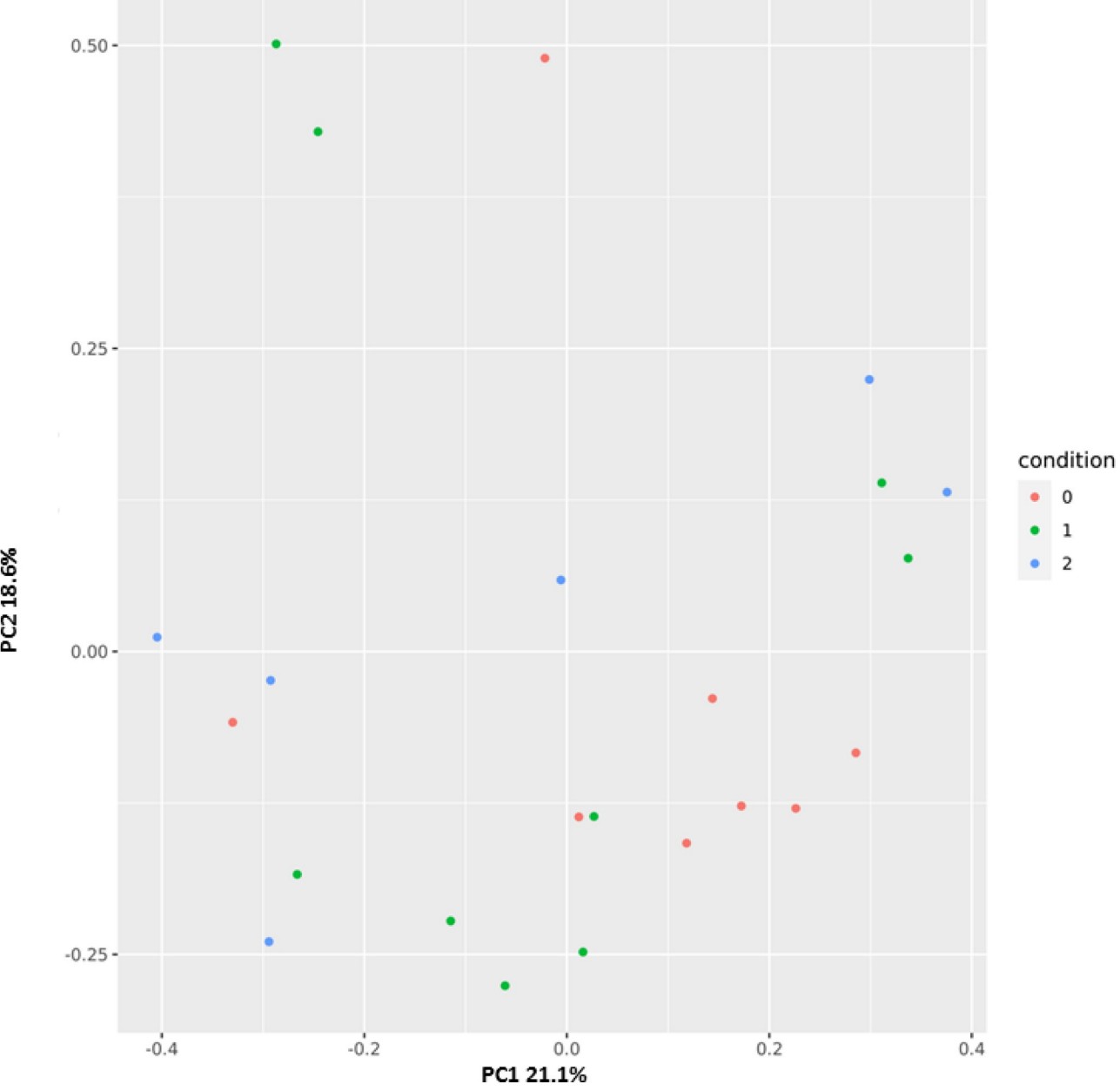

**Fig 2. Principal coordinates plot (PCoA) based on Bray-Curtis distances at family level showing a clustering pattern among samples obtained from controls (red), w-COVID19 (green) and i-COVID19 (blue).**

*Fusobacteria* were decreased (0% vs 0.08% FDR = 0.00 and 0.02% vs 0.04% FDR = 0.00 respectively). When comparing w-COVID19 and i-COVID19 patients' microbiota no significant changes were observed at the Phylum level (Fig 3 and S1–S3 Tables).

At the Family level, a number of potential pathogenic bacteria such as *Peptostreptococcaceae*, *Enterobacteriaceae*, *Staphylococcaceae*, *Vibrionaceae*, *Aerococcaceae*, *Dermabacteraceae*, *Actinobacteria* and others were increased in w-COVID19 as compared to CTRL while *Nitrospiraceae*, *Propionibacteriaceae*, *Aeromonadaceae*, *Moraxellaceae*, *Mycoplasmataceae* were significantly reduced together with others reported in Fig 3 and S4 Table. When considering the i-COVID19 as compared to CTRL, in addition to some bacteria in common with w-COVID19 patients (i.e. *Staphylococcaceae*, *Aerococcaceae*, *Dermabacteraceae*, *Actinobacteria* and so on Fig 4, and Fig 8A and S5 Table) *Erysipelotrichaceae*, *Microbacteriaceae*, *Mycobacteriaceae*, *Pseudonocardiaceae*, *Brevibacteriaceae*, and others reported in S5 Table were significantly increased while Carnobacteriaceae, *Coriobacteriaceae* and *Mycoplasmataceae* were significantly reduced.

Nevertheless *Staphylococcaceae*, *Microbacteriaceae*, *Micrococcaceae*, *Pseudonocardiaceae*, *Erysipelotrichales* and others reported in S6 Table were significantly higher in i-COVID-19 as compared to w-COVID19. *Carnobacteriaceae*, *Pectobacteriaceae*, *Moritellaceae*, *Selenomonadaceae*, *Micromonosporaceae*, *Coriobacteriaceae* and few others were significantly decreased in i-COVID19 as compared to w-COVID19. Individual microbiota profiles are provided in Figs 5 and 6 at the Phylum and Family level.

Moreover, unsupervised hierarchical analysis at the family level (Fig 7) revealed a characteristic microbial signature in CTRL segregated from that one of COVID-19 positive patients. Strikingly, a distinct profile can be distinguished between i-COVID19 and w-COVID19 with the latter being closer to CTRL. Microbiota analysis at lower taxonomic levels (genera and species, reported in S7–S12 Tables).

## 3.4. Differences in microbial populations of w-COVID19 and i-COVID19 patients in comparison to CTRL

At lower taxonomic levels, many differences with CTRL group emerged in both i- and w-COVID19 patients. VENN diagrams showed the number of families (Fig 8A) genera (Fig 8B) and species (Fig 8C) shared or distinctive of the two different groups as compared to CTRL used as reference. Although w-COVID19 and i-COVID19 patients share a number of increased and decreased bacteria, a distinctive bacteria profile can be also observed when compared to CTRL (S13 Table).

## 4. Discussion

It is nowadays well recognized that virus infections can alter the host's microbiota at different sites [18, 19], however is less clear whether changes of microbiota have direct or indirect effects i.e. limiting or promoting viral infections. Microbiota's products such as short chain fatty acids, metabolites or bacteriocine may directly interact with viral particles to alter infectivity or responses to therapy [20, 21].

In our pilot study we demonstrated that SARS-CoV-2 infection is associated with major changes in gut microbiota profile of the patients. The main findings are the reduction of microbial richness in i-COVID19 as compared to CTRL and w-COVID19 indicating that patients in ICU displayed a lower microbial richness as measured by Chao1 index. For the Shannon index the same trend was also observed, but without reaching statistical significance. Our results are in line with those recently obtained by Zuo et al, in which enrichment of opportunistic pathogens and loss of beneficial bacteria was observed [22].

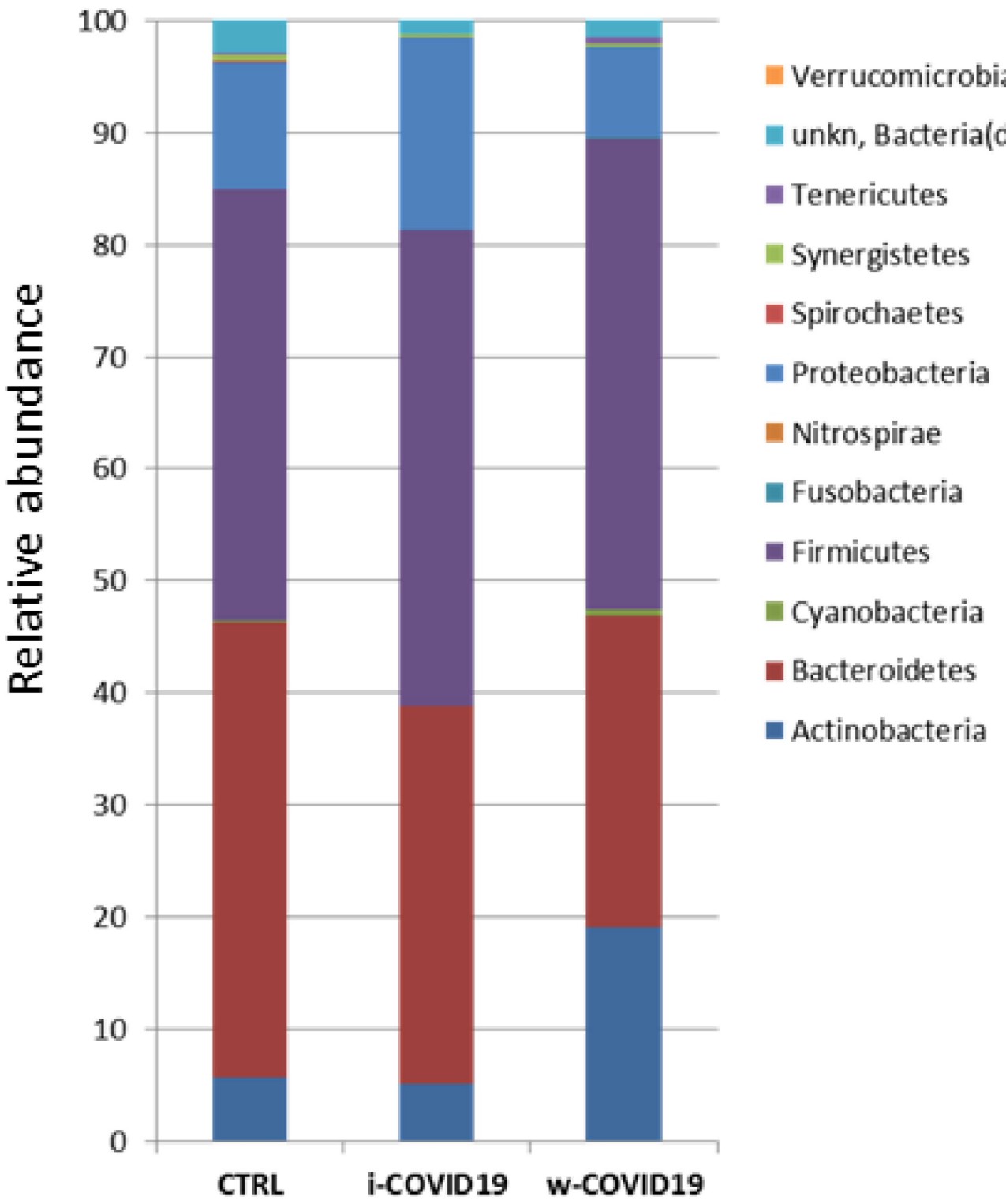

**Fig 3. Microbiota composition of w-COVID19, i-COVID19 patients and CTRL, at the phylum level.** The mean value of all the detected taxa is represented.

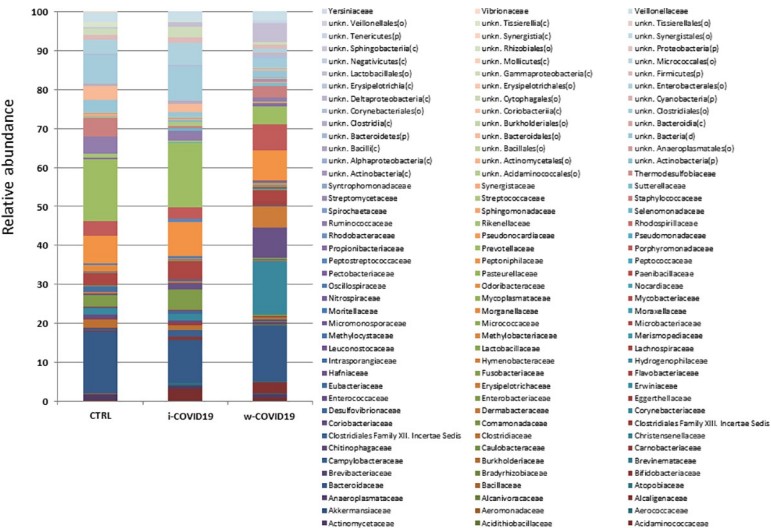

**Fig 4. Microbiota composition of w-COVID19, i-COVID19 patients and CTRL, at the family level.** The mean value of all the detected taxa is represented.

Moreover, ANOVA and Kruskal-Wallis analysis reveal that the variables between our patient cohorts was not noteworthy except for ferritin level that is significantly lower in ICU patients.

Ferritin is a marker of inflammation and the high levels of ferritin detected in i-COVID19 patients in comparison to w-COVID19, may be associate with a greater severity of the disease and adverse outcomes. Normally, ferritin is able to activate macrophages that when stimulated begin to secrete cytokines that at low concentrations, help to protect the body from viruses and bacteria. On the other hand, high levels of ferritin activate more macrophages that produce the so-called "cytokine storm" which can be lethal for the body [23].

As for gut microbiota, antibiotics use can obviously determine a further loss of heterogeneity and composition, leading to down regulation of beneficial symbionts and exacerbation of gut dysbiosis, and for this reason the avoidance of unnecessary antibiotics use in the treatment of viral pneumonia is strongly suggested, as antibiotics can eliminate beneficial bacteria and weaken the gut barrier [24]. Although an increase of pro-inflammatory and potential pathogenic bacteria such as *Peptostreptococcaceae*, *Enterobacteriaceae*, *Staphylococcaceae*, *Vibrionaceae*, *Aerococcaceae*, *Dermabacteraceae*, *Actinobacteria* [25–27], is confirmed in w-COVID19 and i-COVID19 patients, with some of them found in both groups, hierarchical analysis shows a distinct profile between i- and w-COVID19 with the latter being closer to CTRL (Fig 4). Notably, a profound dysbiosis (Fig 5) was observed in one ward patient (90-year-old patient with diabetes, meningioma and osteoporosis in association with an increase of C-Reactive Protein (CRP) and lymphocytes), with a significant increase in *Proteobacteria* and a relevant reduction in *Bacteriodetes*, reflecting an important inflammatory state. Growing evidence has shown that perturbation of the gut microbial community may fuel blooms of otherwise low abundance and harmful bacteria which can further exacerbate the intestinal inflammation. Indeed, dysbiosis in the distal gut is often characterized by a decrease in the prevalence of strict anaerobes and an increased relative abundance of facultative anaerobic bacteria.

This could also contribute to the lower severity of symptoms of w-COVID19 as compared to patients admitted to ICU. This evidence is confirmed by our three CTRL patients hospitalzed in ICU that had a nasopharyngeal PCR negative for SARS-CoV2. The latter, showed an

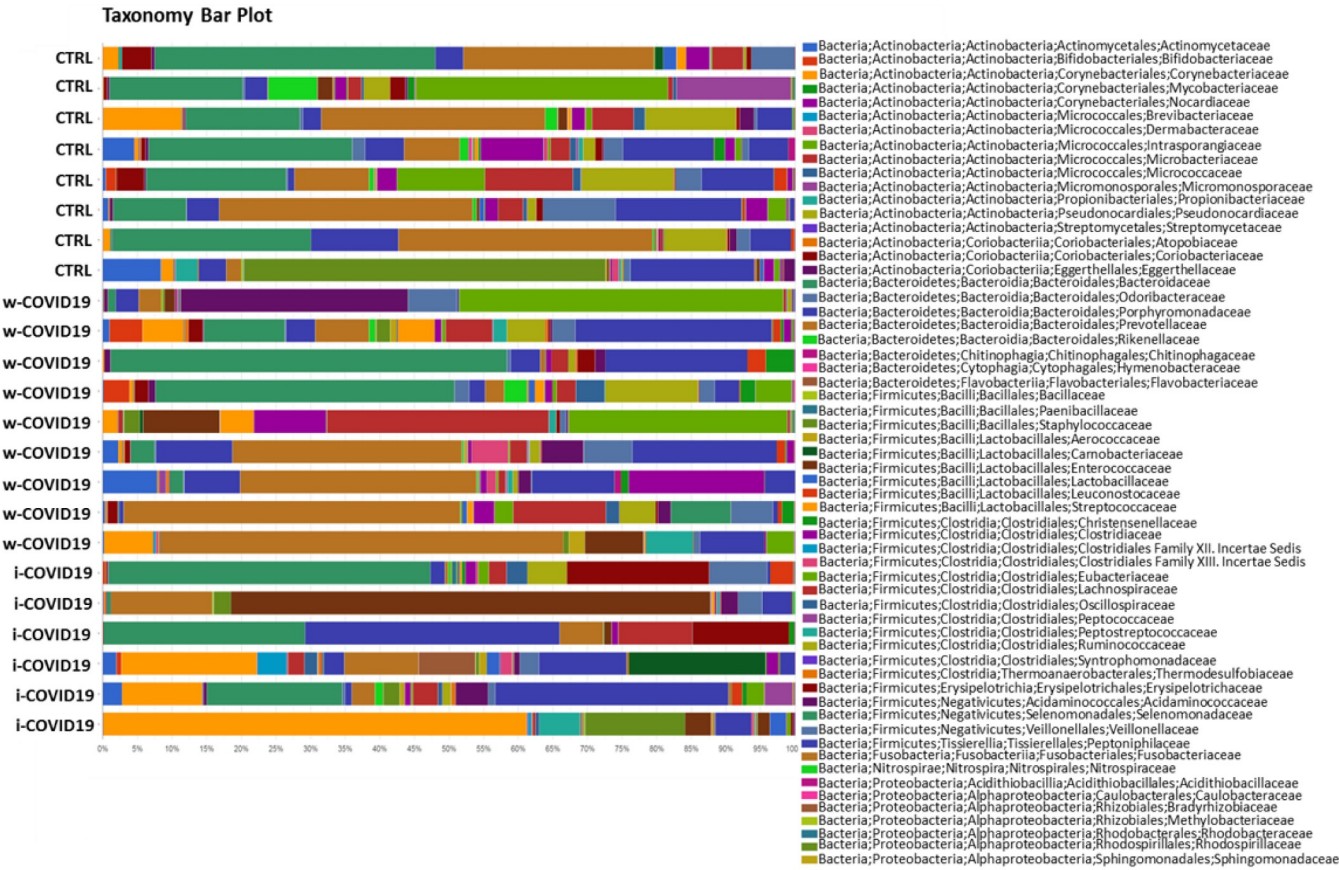

**Fig 5. Taxonomy bar plot showing the individual microbiota profile at phylum levels.**

increase in *Erysipelotrichaceae* which are involved in inflammation-related disorders of the GI tract [28]. These data are in agreement with a previous study [29] where *Erysipelotrichaceae* were found associated with COVID-19 severity. Noteworthy a strong decrease of

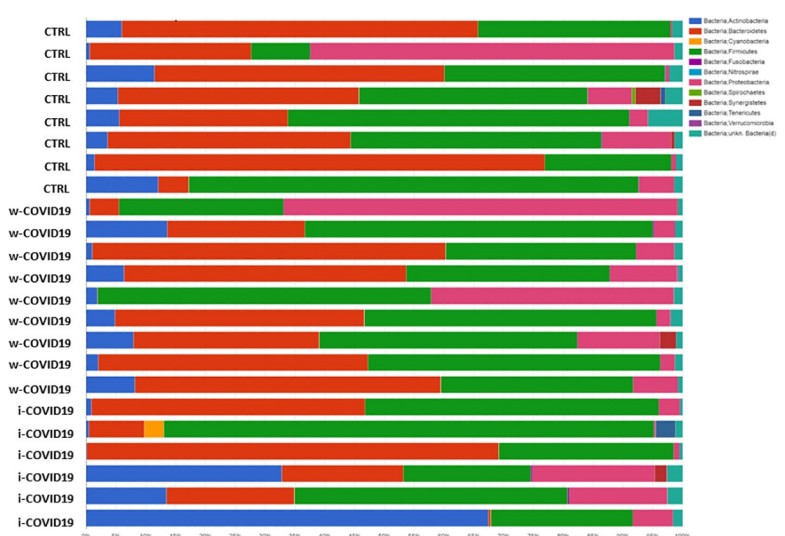

**Fig 6. Taxonomy bar plot showing the individual microbiota profile at family levels.**

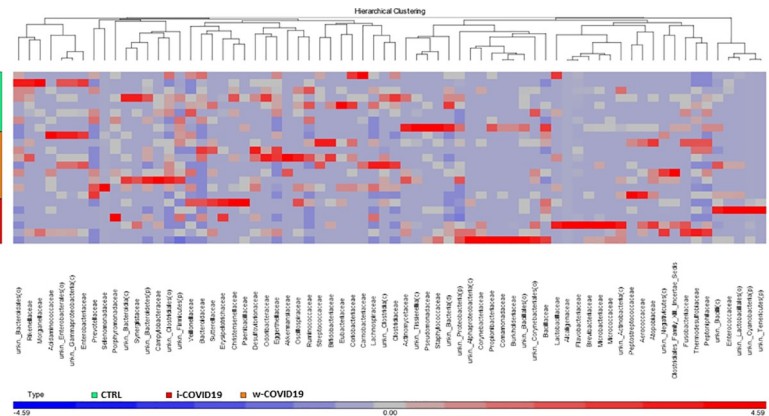

**Fig 7. Heatmap of one-way hierarchical clustering of differentially abundant families among the three cohorts.** A dual-color code counts for species up- (red) and down-represented (blue), respectively.

*Faecalibacterium* (which is associated to Crohn's disease, colonrectal tumor, diabetes and non alcoholic steatohepatitis) [30] in i-COVID19 patients was detected together with a reduction of *Ruminococcaceae*, *Clostridiaceae* which are involved in Short Chain Fatty Acids (SCFAs) production among which butyrate presents potent antinflammatory properties [31]. Among the species (S9 Table), it is worth of note the decrease of *Bacteroides dorei*, *Bacteroides thetaiotaomicron* in w-COVID19 as compared to CTRL which are known to down-regulate ACE2 expression in the murine gut [19].

There are several limitations to this study. First, this is a pilot study conducted in a single center in urban area in central Italy with a limited number of enrolled patients. Our preliminary observations on the likely impact of SARS-CoV-2 infection on gut microbiota need to be confirmed in larger comparative trial including paucysimptomatic or asymptomatic COVID-

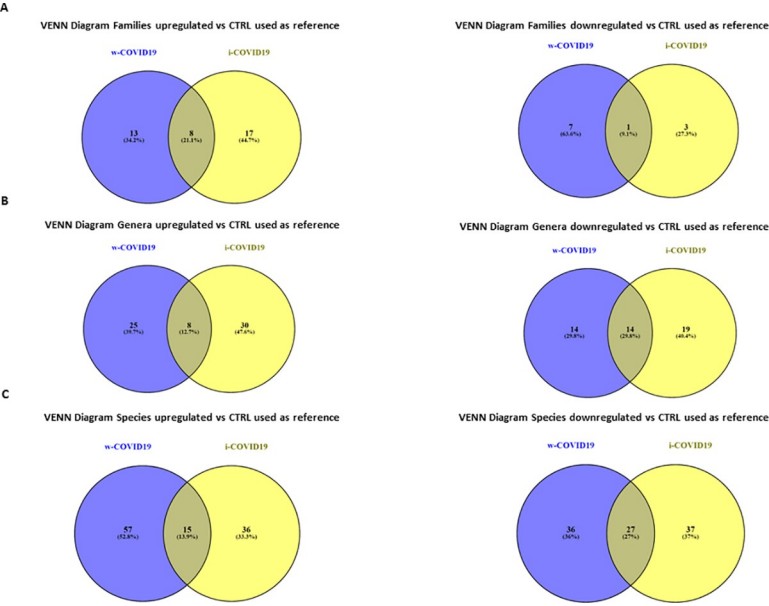

**Fig 8.** Venn diagrams showing the number of distinct and shared families (A), genera (B) and species (C) up and decreased between subjects grouped by w-COVID19, i-COVID19 patients as compared to CTRL used as referenced.

19 patients a part from those admitted with severe or critical disease. Moreover, another limit of our work was related to our ICU patients; as reported in several papers in literature, the long stay of ICU patients can change the microbiota gut composition [32]. Definitely more patients should be included to evaluate the microbiota composition in non-COVID-19 ICU patients in comparison with w-COVID19 and i-COVID19. However, among our CTRL, three subjects were non-COVID-19 ICU patients and rectal swabs were performed one or two days after the hospitalization. Furthermore, the antibiotic treatment for ICU patients could have affected the microbiota profile, but the antibiotic therapy was administered a few days after the hospitalization in ICU, so could be assumed that this effect may not be significant. As for the link between clinical data and microbiota profiles, we are aware that age, gender and co-morbidities are factors that strongly influence the microbiota profiles, however in our study population these demographic features were not statistically significant different among the three groups.

The use of rectal swabs for gut microbiota analysis instead of standard fecal samples is not fully endorsed. Currently, feces or mucosal biopsy specimens are the biological samples most commonly used for standard 16S analysis [33]. However, as shown in several studies, stool and rectal swab are highly similar, indicating that these sampling methods could be used interchangeably to assess the community structure of the distal GI tract [34, 35]. Finally, no data on the molecular detection of SARS-CoV-2 in rectal swabs are presented in our patients. Current knowledge on whether fecal transmissibility (either orally, through fomites, or by aspiration of fecal contaminated droplets) is likely to be an important mode of COVID-19 transmission, is still limited [36]. Although out of our objective, this is an interesting research topic, particularly in health care facilities with incontinent residents.

In conclusion, significant changes in gut microbial communities in patients with COVID-19 pneumonia with different disease severity compared to CTRL have been identified. Specific microbial signatures in COVID-19 patients and roles of gut microbiota in different phase of disease and hospital setting are needed to be investigated and validated in larger cohorts.

## Supporting information

**S1 Table. Deseq2_phylum level_w-COVID19 vs CTRL used as reference.**
(XLSX)

**S2 Table. Deseq2_phylum level_i-COVID19 vs CTRL used as reference.**
(XLSX)

**S3 Table. Deseq2_phylum level_i-COVID19 vs w-COVID19 used as reference.**
(XLSX)

**S4 Table. Deseq2_family level_w-COVID19 vs CTRL used as reference.**
(XLSX)

**S5 Table. Deseq2_family level_i-COVID19 vs CTRL used as reference.**
(XLSX)

**S6 Table. Deseq2_family level_i-COVID19 vs w-COVID19 used as reference.**
(XLSX)

**S7 Table. Deseq2_genus level_w-COVID19 vs CTRL used as reference.**
(XLSX)

**S8 Table. Deseq2_genus level_i-COVID19 vs CTRL used as reference.**
(XLSX)

**S9 Table. Deseq2_genus level_i-COVID19 vs w-COVID19 used as reference.**
(XLSX)

**S10 Table. Deseq2_species level_w-COVID19 vs CTRL used as reference.**
(XLSX)

**S11 Table. Deseq2_species level_i-COVID19 vs CTRL used as reference.**
(XLSX)

**S12 Table. Deseq2_specie level_i-COVID19 vs w-COVID19 used as reference.**
(XLSX)

**S13 Table. Shared and distinct bacteria in w-COVID19 and i-COVID19 versus CTRL used as reference.**
(XLSX)

## Acknowledgments

**The authors gratefully acknowledge the Collaborators Members of the National Institute for Infectious Diseases (INMI) COVID-19 study group:** Maria Alessandra Abbonizio, Amina Abdeddaim, Chiara Agrati, Fabrizio Albarello, Gioia Amadei, Alessandra Amendola, Mario Antonini, Andrea Antinori, Tommaso Ascoli Bartoli, Francesco Baldini, Raffaella Barbaro, Barbara Bartolini, Rita Bellagamba, Martina Benigni, Nazario Bevilacqua, Gianlugi Biava, Michele Bibas, Licia Bordi, Veronica Bordoni, Evangelo Boumis, Marta Branca, Donatella Busso, Marta Camici, Paolo Campioni, Maria Rosaria Capobianchi, Alessandro Capone, Cinzia Caporale, Emanuela Caraffa, Ilaria Caravella, Fabrizio Carletti, Concetta Castilletti, Adriana Cataldo, Stefano Cerilli, Carlotta Cerva, Roberta Chiappini, Pierangelo Chinello, Carmine Ciaralli, Stefania Cicalini, Francesca Colavita, Angela Corpolongo, Massimo Cristofaro, Salvatore Curiale, Alessandra D'Abramo, Cristina Dantimi, Alessia De Angelis, Giada De Angelis, Maria Grazia De Palo, Federico De Zottis, Virginia Di Bari, Rachele Di Lorenzo, Federica Di Stefano, Gianpiero D'Offizi, Davide Donno, Francesca Faraglia, Federica Ferraro, Lorena Fiorentini, Andrea Frustaci, Matteo Fusetti, Vincenzo Galati, Roberta Gagliardini, Paola Gallì, Gabriele Garotto, Saba Gebremeskel Tekle, Maria Letizia Giancola, Filippo Giansante, Emanuela Giombini, Guido Granata, Maria Cristina Greci, Elisabetta Grilli, Susanna Grisetti, Gina Gualano, Fabio Iacomi, Giuseppina Iannicelli, Giuseppe Ippolito, Eleonora Lalle, Simone Lanini, Daniele Lapa, Luciana Lepore, Raffaella Libertone, Raffaella Lionetti, Giuseppina Liuzzi, Laura Loiacono, Andrea Lucia, Franco Lufrani, Manuela Macchione, Gaetano Maffongelli, Alessandra Marani, Luisa Marchioni, Andrea Mariano, Maria Cristina Marini, Micaela Maritti, Alessandra Mastrobattista, Giulia Matusali, Valentina Mazzotta, Paola Mencarini, Silvia Meschi, Francesco Messina, Annalisa Mondi, Marzia Montalbano, Chiara Montaldo, Silvia Mosti, Silvia Murachelli, Maria Musso, Emanuele Nicastri, Pasquale Noto, Roberto Noto, Alessandra Oliva, Sandrine Ottou, Claudia Palazzolo, Emanuele Pallini, Fabrizio Palmieri, Carlo Pareo, Virgilio Passeri, Federico Pelliccioni, Antonella Petrecchia, Ada Petrone, Nicola Petrosillo, Elisa Pianura, Carmela Pinnetti, Maria Pisciotta, Silvia Pittalis, Agostina Pontarelli, Costanza Proietti, Vincenzo Puro, Paolo Migliorisi Ramazzini, Alessia Rianda, Gabriele Rinonapoli, Silvia Rosati, Martina Rueca, Alessandra Sacchi, Alessandro Sampaolesi, Francesco Sanasi, Carmen Santagata, Alessandra Scarabello, Silvana Scarcia, Vincenzo Schininà, Paola Scognamiglio, Laura Scorzolini, Giulia Stazi, Fabrizio Taglietti, Chiara Taibi, Roberto Tonnarini, Simone Topino, Francesco Vaia, Francesco Vairo, Maria Beatrice Valli, Alessandra Vergori, Laura Vincenzi, Ubaldo Visco-Comandini, Serena Vita, Pietro Vittozzi, and Mauro Zaccarelli.

## Author Contributions

**Conceptualization:** Emanuele Nicastri, Valerio Pazienza.

**Formal analysis:** Antonio Mazzarelli, Maria Letizia Giancola, Anna Farina, Cesare Ernesto Maria Gruber, Tommaso Ascoli Bartoli, Valerio Pazienza.

**Funding acquisition:** Barbara Bartolini.

**Investigation:** Antonio Mazzarelli, Maria Letizia Giancola.

**Methodology:** Antonio Mazzarelli, Luisa Marchioni, Martina Rueca.

**Supervision:** Barbara Bartolini, Gaetano Maffongelli, Maria Rosaria Capobianchi, Giuseppe Ippolito, Antonino Di Caro.

**Writing – review & editing:** Antonio Mazzarelli, Maria Letizia Giancola, Antonino Di Caro, Emanuele Nicastri, Valerio Pazienza.

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
