## [Decision Letter · Decision Letter 0]

17 Nov 2020

PONE-D-20-29220

16S rRNA Gene Sequencing of Rectal Swab in Patients Affected by COVID-19

PLOS ONE

Dear Dr. Di Caro,

Thank you for submitting your manuscript to PLOS ONE. After careful consideration, we feel that it has merit but does not fully meet PLOS ONE’s publication criteria as it currently stands. Therefore, we invite you to submit a revised version of the manuscript that addresses the points raised during the review process.

The reviewers have raised several concerns that will need to be fully addressed in your revised manuscript. 

We look forward to receiving your revised manuscript.

Kind regards,

Jane Foster, PhD

Academic Editor

PLOS ONE

Journal Requirements:

2)  Thank you for stating the following financial disclosure:

 [NO-The funders had no role in study design, data collection and analysis, decision to publish, or preparation of the manuscript.].

3) Thank you for stating the following in your Competing Interests section: 

[NO-authors have competing interests].

4) In your Data Availability statement, you have not specified where the minimal data set underlying the results described in your manuscript can be found. PLOS defines a study's minimal data set as the underlying data used to reach the conclusions drawn in the manuscript and any additional data required to replicate the reported study findings in their entirety. All PLOS journals require that the minimal data set be made fully available. For more information about our data policy, please see http://journals.plos.org/plosone/s/data-availability.

5) Please upload a new copy of Figures 4 and 5 as the detail is not clear. Please follow the link for more information: https://blogs.plos.org/plos/2019/06/looking-good-tips-for-creating-your-plos-figures-graphics/

Reviewers' comments:

Reviewer's Responses to Questions

**Comments to the Author**

1. Is the manuscript technically sound, and do the data support the conclusions?

Reviewer #1: Yes

Reviewer #2: No

2. Has the statistical analysis been performed appropriately and rigorously? 

Reviewer #1: I Don't Know

Reviewer #2: I Don't Know

3. Have the authors made all data underlying the findings in their manuscript fully available?

Reviewer #1: Yes

Reviewer #2: Yes

4. Is the manuscript presented in an intelligible fashion and written in standard English?

Reviewer #1: Yes

Reviewer #2: Yes

5. Review Comments to the Author

Reviewer #1: The authors have performed a pilot study examining the gut microbiomes of patients admitted to the ICU or infectious disease ward at their institution with or without COVID19-associated pneumonia. The data support a loss of bacterial diversity in COVID-positive patients and gain of potentially pathological species.

Revisions I think should be made before publication include:

1) The methods are too vague.

a. For example, was the PCOA performed in GAIA? What about the Venn Diagrams, etc.? It's not stated, nor for most of the other figures.

b. How were the different bacterial profiles generated by the different hypervariable regions combined? How did they differ?

c. How many reads were generated per patient? Was this sufficient to capture the long tail of the gut microbiomes?

d. What was the similarity level used to bin reads and identify OTUs? If this was all done using default settings in GAIA, it should stated.

e. One hypervariable region is listed as V-2-4-8 which does not make sense, I think they should be listed as V2, V4, V8 to differentiate these single hypervariable region amplicons from the V3-6 amplicons which actually span hypervariable regions V3 through V6.

f. which NCBI database was used?

2) The results lack quantitation. For example in lines 158-176 no fold changes in relative abundance are reported for the organisms listed. Nor is there discussion of the prevalence of these bacteria that may constitute potential biomarkers of COVID infection. For example, In supplemental table 1, Spirochaetes and Fusobacteria were found to be significantly down in ward patients relative to control, but neither Phylum is prevalent enough to even be visible in Figure S1. Are these results just noise from very rare organisms that may or may not be found at the sampling depth of the study? While the statistics are this is not the case, and the authors have adjusted for multiple testing (FDR p-values) some discussion of the low prevalence of the significantly altered organisms would make the findings more convincing.

Another way to look at my concern here is to consider why it is the ward and ICU patients have differences in their affected bacterial families in lines 158-168 (while also having similarities) when both cohorts have COVID infection, If the families listed were important to COVID infection then they should be found in both ward and ICU patients. Either there is an affect of ward versus ICU which should be discussed or these families are noise in the data despite passing statistical muster.

3) There appears to be significant variation between patients within the three main categories (ward, ICU, control). For example, in supplementary figure 1 there is one control and one ward patient with surprisingly high Proteobacteria and low Firmicutes/Bacteriodetes. In addition, one ward patient apparently has no Bacteriodetes at all - a shocking result given this is typically the major Phylum on gut microbiomes. There should be some discussion of the variation and it's potential to affect the development of potential biomarkers.

4) Some discussion of why there was a significant difference in Ferritin in the ICU patients would be helpful for a non-clinical audience.

Minor changes:

1) Figure S1 the type is difficult to read because of low resolution, try to use vector type rather than rasterized type.

2) Figure S2 the legend is cut of on the right side making it impossible, for example, to identify the Fusobacteria. The type is also nearly unreadable due to rasterization.

3)Figure 5 bacterial names are unreadable due to low resolution.

4) line 36 replace remarkably with remarkable

5) line 46 replace different with several

6) line 67 replace examined with performed

7) line 138 replace resulted with was

8) line 150, 155 insert the between At and Phylum

9) line 150,153 replace resulted with were

10) line 158 insert the between At and Family

11) line 218, 225 replace decreased with decrease

12) line 229 the phrase "it popped out" is awkward and not formal English for a publication

13) line 232 the first sentence would read better if simply changed to "There are several limitations to this study."

14) insert a between administered and few

15) line 247 replace study with studies

Reviewer #2: PONE-D-20-29220 16S rRNA Gene Sequencing of Rectal Swab in Patients Affected by COVID-19

In the current study, the Researchers investigated whether COVID infection promoted alterations in the gut microbiome diversity and profiles, as well as whether these shifts conferred severity of the disease state. While the study is limited in subject enrollment (which can be overlooked since it is a pilot study), there are some potentially novel outcomes of the data sets provided. However, due to the vast limitations of the study, further details on methodologies and additional analyses are required. Furthermore, the authors should temper their conclusions drawn from the current data set, as the study design does not allow for determination that COVID infection mediated or promoted the observed outcomes.

Major concerns/comments:

1. The major concern of this Reviewer is that there is no "pre-COVID" data on the gut microbiome profiles of these patients, so it is impossible to definitively conclude that these microbiota profiles are due to the COVID infection. These profiles could be due to variation in diets, environment, medications, and/or existing comorbidities. At the very least this limitation needs to be addressed and the conclusions that these data provide evidence for COVID mediating these outcomes needs to be tempered.

2. The individual microbiota graphs need to be provided in the main text vs. as supplementary files. Furthermore, it appears that for most statistical endpoints reported in the group averages, only 1 or 2 individuals in each group drove those changes represented in the averages. For example Proteobacteria at the Phylum level are so varied, even amongst the controls, that it appears that the averages presented in Figures 3 and 4 are not necessarily representative of the whole group.

3. In addition, information for each of the subjects should be correlated to their individual microbiome - this can be provided in a table in the supplemental files (e.g. Subject Control 1, age, sex, antibiotics, comorbidities). This is important, because it may allow for further distinction of effects of antibiotics and co-morbidities on the highly varied individual microbiome reads.

4. Further clarification on the analysis of the 16S sequencing needs to be provided. What were total reads, and were they comparable across all samples? Was the data rarefied for alpha and beta diversity analysis?

5. A more in-depth description of how samples were collected needs to be provided. One of the main concerns with microbiome analysis is minimizing contamination from other sources in your samples, which in this case would have been difficult. As such, can the authors rule out that the microbiota analyses do not include those microbiota on the surface of the skin vs. the actual GI tract? This is one of the main downfalls of using a rectal swab vs. fecal samples.

6. Were the patients that tested negative for COVID-19 (the controls) only tested 1 time?

7. Is the PCoA graph presented in Fig 2 weighted or unweighted?

8. Analysis of Molecular Variance (AMOVA) and Analysis of similarities (ANOSIM) to assess the variations and similarities among different groups would be beneficial, especially considering the PCoA graph does not show distinct clustering for most groups.

9. Additional alpha diversity analyses should be completed since only one of the chosen tests revealed statistical significance. The authors may want to consider abundance coverage-based estimator (ACE) richness and evenness analyses.

6. PLOS authors have the option to publish the peer review history of their article (what does this mean?). If published, this will include your full peer review and any attached files.

Reviewer #1: No

Reviewer #2: No

---

## [Author Response · Author response to Decision Letter 0]

3 Dec 2020

Review Comments to the Author

Reviewer #1: The authors have performed a pilot study examining the gut microbiomes of patients admitted to the ICU or infectious disease ward at their institution with or without COVID19-associated pneumonia. The data support a loss of bacterial diversity in COVID-positive patients and gain of potentially pathological species.

Revisions I think should be made before publication include:

1) The methods are too vague.

a. For example, was the PCOA performed in GAIA? What about the Venn Diagrams, etc.? It's not stated, nor for most of the other figures.

RE: We thank the reviewer for this comment and we apologize for the missing information. We have now provided more details about the methods in the respective paragraph. 

b. How were the different bacterial profiles generated by the different hypervariable regions combined? How did they differ?

RE: We apologize for the lack of clarity. The sequencing run has generated in total 16x10^6 reads with the 77% of high quality reads (21% low quality, 2% test fragments). Finally we obtained 5.5x10^5 reads per sample (reads length were 244 bp mean, 260 bp median and 289 bp mode) and all analyzed specimens showed a suitable library’s profile. The analysis was performed by 16S Metagenomics GAIA 2.0 software and DESeq2 package software. Sequence data generated as FASTQ files, were analysed using the 16S Metagenomics GAIA 2.0 software which performs the quality control of the reads/pairs (i.e., trimming, clipping and adapter removal steps) through FastQC and BBDuk. The reads/pairs are mapped with BWA-MEM against the 16S databases (based on NCBI). Differential expression analysis using DESeq2 package to test for differential analysis by use of negative binomial generalized linear models was used. Only changes with FDR below 0.05 were considered significant.

c. How many reads were generated per patient? Was this sufficient to capture the long tail of the gut microbiomes?

RE: We thank the reviewer for this comment. We have obtained an average of 5.5 x 10^5 reads each sample. Reads were sufficient to evaluate optimal rarefaction curves as shown in the following graphics. Graphics were plotted based on Chao index, observed species, shannon and simpson index respectively. The rarefaction curves suggest that the samples have plateaued; this means that a good representation of the microbial community was obtained for the samples analysed in this study.

d. What was the similarity level used to bin reads and identify OTUs? If this was all done using default settings in GAIA, it should stated.

RE: All analyses are executed with default settings described in https://www.biorxiv.org/content/10.1101/804690v1. The analysis is amplicon-seq, so the identity threshold for species is 97%, for genus is 93%.

e. One hypervariable region is listed as V-2-4-8 which does not make sense, I think they should be listed as V2, V4, V8 to differentiate these single hypervariable region amplicons from the V3-6 amplicons which actually span hypervariable regions V3 through V6.

RE: We agree with the reviewer. We have now added these changes within the text.

f. which NCBI database was used?

RE: RE: 16S sequences available from GenBank with meaningful taxonomic information in at least the genus level (those classified as uncultured, unidentified, etc. were excluded). In addition, 16S sequences were rescued from entire genomes that also available from GenBank using the corresponding annotation.

2) The results lack quantitation. For example in lines 158-176 no fold changes in relative abundance are reported for the organisms listed. Nor is there discussion of the prevalence of these bacteria that may constitute potential biomarkers of COVID infection. For example, In supplemental table 1, Spirochaetes and Fusobacteria were found to be significantly down in ward patients relative to control, but neither Phylum is prevalent enough to even be visible in Figure S1. Are these results just noise from very rare organisms that may or may not be found at the sampling depth of the study? While the statistics are this is not the case, and the authors have adjusted for multiple testing (FDR p-values) some discussion of the low prevalence of the significantly altered organisms would make the findings more convincing.

RE: We thank the reviewer for this comment. Since numerous changes were observed, we preferred to report p values, FDR score and fold changes in the supplementary tables instead of reporting these values within the manuscript which would have been difficult to read with all the number inserted. We have now added more sentences in order to better describe the changes observed tanking in consideration also the comment below.

Another way to look at my concern here is to consider why it is the ward and ICU patients have differences in their affected bacterial families in lines 158-168 (while also having similarities) when both cohorts have COVID infection, If the families listed were important to COVID infection then they should be found in both ward and ICU patients. Either there is an affect of ward versus ICU which should be discussed or these families are noise in the data despite passing statistical muster.

RE: We thank the reviewer for this insightful comment. As stated in our manuscript, this is a pilot study that represent preliminary data and we are aware that the patient’s number is limited. We cannot exclude that increasing the patients number to be enrolled it is possible to discover more families having similarities in both groups. Nevertheless, several families in common were linked to the virus presence independently of the ward in which the patients are hospitalized. Furthermore, the severity disease and the intestinal inflammation state were different so the variability expressed at the family level is higher in ICU patients in comparison to those observed in patients hospitalized in the ward as also reported by Ravi et al (PMID: 31526447). Our intent was to describe the common and the exclusive microorganisms belonging to the different groups. We agree that families listed due to only COVID infection (either ward or ICU) are worth of note and for this reason we reported that when considering the i-COVID19 as compared to CTRL, in addition to some bacteria in common with w-COVID19 patients (i.e. Staphylococcaceae, Aerococcaceae, Dermabacteraceae, Actinobacteria and so on Fig.4, and Fig. 6A and Supporting material S5 table) Erysipelotrichaceae, Microbacteriaceae, Mycobacteriaceae, Pseudonocardiaceae, Brevibacteriaceae, and others reported in Supporting material S5 table were significantly increased while Carnobacteriaceae, Coriobacteriaceae and Mycoplasmataceae were significantly reduced. 

But we also underlined the exclusive microorganisms belonging to the two distinct groups: Staphylococcaceae, Microbacteriaceae, Micrococcaceae, Pseudonocardiaceae, Erysipelotrichales and others reported in Supporting material S6 table were significantly higher in i-COVID-19 as compared to w-COVID19. Carnobacteriaceae, Pectobacteriaceae, Moritellaceae, Selenomonadaceae, Micromonosporaceae, Coriobacteriaceae and few others were significantly decreased in i-COVID19 as compared to w-COVID19. 

We then discussed all these changes considering the already known microorganisms reported within the literature.

3) There appears to be significant variation between patients within the three main categories (ward, ICU, control). For example, in supplementary figure 1 there is one control and one ward patient with surprisingly high Proteobacteria and low Firmicutes/Bacteriodetes. In addition, one ward patient apparently has no Bacteriodetes at all - a shocking result given this is typically the major Phylum on gut microbiomes. There should be some discussion of the variation and it's potential to affect the development of potential biomarkers.

RE: We thank the reviewer for this comment. We noticed also these changes and we discussed what we thought were the most important. However, in agreement with the reviewer comment we have now added some discussion on the above mentioned changes.

4) Some discussion of why there was a significant difference in Ferritin in the ICU patients would be helpful for a non-clinical audience.

RE: We thank the reviewer for this comment. We better clarified the concept with the following sentences reported in the manuscript: ‘Moreover, ANOVA and Kruskal-Wallis analysis reveal that the variables between our patient cohorts was not noteworthy except for ferritin level that is significantly lower in ICU patients. 

Ferritin is a marker of inflammation and the high levels of ferritin detected in i-COVID19 patients in comparison to w-COVID19, may be associate with a greater severity of the disease and adverse outcomes. Normally, ferritin is able to activate macrophages that when stimulated begin to secrete cytokines that at low concentrations, help to protect the body from viruses and bacteria. On the other hand, high levels of ferritin activate more macrophages that produce the so-called "cytokine storm" which can be lethal for the body’ (PMID: 32268212).

Minor changes:

1) Figure S1 the type is difficult to read because of low resolution, try to use vector type rather than rasterized type.

RE: We regret that the figures are not visible, we have carefully checked all of the figures and the resolution is 300 DPI or more, as requested, we hope to the reviewer’s satisfaction. We trust that we have now resolved the issues pertaining to poor resolution, which can be also resolved by accessing the access/download high-resolution link for each image provided within the text

2) Figure S2 the legend is cut of on the right side making it impossible, for example, to identify the Fusobacteria. The type is also nearly unreadable due to rasterization.

RE: We trust that we have now resolved the issues

3) Figure 5 bacterial names are unreadable due to low resolution.

RE: We regret that the figures are not visible, we have carefully checked all of the figures and the resolution is 300 DPI or more, as requested, we hope to the reviewer’s satisfaction. We trust that we have now resolved the issues pertaining to poor resolution which can be also resolved by accessing the access/download high-resolution link for each image provided within the text

4) line 36 replace remarkably with remarkable; 5) line 46 replace different with several; 6) line 67 replace examined with performed; 7) line 138 replace resulted with was; 8) line 150, 155 insert the between At and Phylum; 9) line 150,153 replace resulted with were; 10) line 158 insert the between At and Family; 11) line 218, 225 replace decreased with decrease;12) line 229 the phrase "it popped out" is awkward and not formal English for a publication; 13) line 232 the first sentence would read better if simply changed to "There are several limitations to this study."; 14) insert a between administered and few; 15) line 247 replace study with studies.

RE: We thank the reviewer for these suggestions. We have now edited the text according to the reviewer’ comments.

---

## [Decision Letter · Decision Letter 1]

5 Jan 2021

PONE-D-20-29220R1

16S rRNA Gene Sequencing of Rectal Swab in Patients Affected by COVID-19

PLOS ONE

Dear Dr. Di Caro,

Thank you for submitting your manuscript to PLOS ONE. After careful consideration, we feel that it has merit but does not fully meet PLOS ONE’s publication criteria as it currently stands. Therefore, we invite you to submit a revised version of the manuscript that addresses the points raised during the review process.

We look forward to receiving your revised manuscript.

Kind regards,

Jane Foster, PhD

Academic Editor

PLOS ONE

Reviewers' comments:

Reviewer's Responses to Questions

**Comments to the Author**

1. If the authors have adequately addressed your comments raised in a previous round of review and you feel that this manuscript is now acceptable for publication, you may indicate that here to bypass the “Comments to the Author” section, enter your conflict of interest statement in the “Confidential to Editor” section, and submit your "Accept" recommendation.

Reviewer #1: (No Response)

Reviewer #2: (No Response)

2. Is the manuscript technically sound, and do the data support the conclusions?

Reviewer #1: Yes

Reviewer #2: Partly

3. Has the statistical analysis been performed appropriately and rigorously? 

Reviewer #1: Yes

Reviewer #2: I Don't Know

4. Have the authors made all data underlying the findings in their manuscript fully available?

Reviewer #1: Yes

Reviewer #2: Yes

5. Is the manuscript presented in an intelligible fashion and written in standard English?

Reviewer #1: Yes

Reviewer #2: Yes

6. Review Comments to the Author

Reviewer #1: The authors have thoughtfully addressed most of the concerns of both reviewers, however, they have done so in the response to reviewers, and not in the manuscript. For example, both reviewers noted that the number of reads per sample was not reported which goes a long way to assuaging a knowledgeable reader's concerns about sequencing depth, especially when paired with the alpha diversity analysis also shown only in response to reviewers. The authors responded with the details that should be in the methods, but did not add them to the manuscript. Specifically to the point above, something like the following text:

"The sequencing run has generated in total 16x10^6 reads with the

77% of high quality reads(21% low quality, 2% test fragments). Finally we obtained 5.5x10^5 reads per sample

(reads length were 244 bp mean, 260 bp median and 289 bp mode) and all analyzed specimens showed a

suitable library’s profile. The analysis was performed by 16S Metagenomics GAIA 2.0 software and DESeq2

package software. Sequence data generated as FASTQ files, were analysed using the 16S Metagenomics GAIA

2.0 software which performs the quality control of the reads/pairs (i.e., trimming, clipping and adapter

removal steps) through FastQC and BBDuk. The reads/pairs are mapped with BWA-MEM against the 16S

databases (based on NCBI). Differential expression analysis using DESeq2 package to test for differential

analysis by use of negative binomial generalized linear models was used. Only changes with FDR below 0.05

were considered significant"

should be added to the methods, not just in the response to reviewers. Think of the reviewers as interested readers, if we both wanted this information, so will the readers. Similarly, the percent similarity used to determine species and genus calls, details on the reference sequence database used, etc. should be in the methods. In addition, it would be an improvement to include the CHAO1 alpha diversity plot in the response to reviewers in the supplemental figures. There is no word limit or figure limit in PLOS ONE, so there is no reason to leave out these details which provide the reader with key information to understand and trust the results.

In addition, it is still unclear how the analysis was performed on the reads, while the methods paragraph quoted above would suffice, later in the response to reviewers (reviewer 2, point 9) Ion Reporter is mentioned and strongly implies the reads were processed in some way on that platform. However, no mention of Ion Reporter is made in the manuscript. In addition, use of Ion Reporter would explain how the identity calls made from the multiple hypervariable regions were combined, but the authors still have not addressed this question in the response to reviewers or the manuscript (reviewer 1, point 1b).

Lastly, In response to reviewer 1, point 3 the authors state that discussion of the strikingly absence of Bacteriodetes in one patient has been added to the text, however, this reviewer cannot find it in the revised manuscript and a search of the revised manuscript for the word "Bacteriodetes" returned no results. If the text has indeed been added, the authors should refer to specific line numbers.

Reviewer #2: While the authors have addressed many of the initial concerns in the revised manuscript, there are still some issues that need to be further addressed (some of which were in the initial major concerns/comments previously):

1. The Supplemental Figures S1 and S2 need to be moved to the main body of the manuscript, instead of being placed in the Supplemental files. These are extremely important figures for transparency of the major limitations of this study (small sample size, antibiotic use, no pre- vs. post data, etc.). The reader should not have to seek this information in the Supplemental Files, since they pertain to the main findings of the study, especially since individual variation seemed to drive the significant response in some of the reported outcomes within study groups.

2. The authors should address in the Discussion (at a minimum) whether age, gender, co-morbidities, and/or antibiotic treatment (n=11 on study) altered individual 16S microbiota profiles reported.

7. PLOS authors have the option to publish the peer review history of their article (what does this mean?). If published, this will include your full peer review and any attached files.

Reviewer #1: **Yes: **George S. Watts

Reviewer #2: No

---

## [Author Response · Author response to Decision Letter 1]

8 Jan 2021

Reviewer #1: The authors have thoughtfully addressed most of the concerns of both reviewers, however, they have done so in the response to reviewers, and not in the manuscript. For example, both reviewers noted that the number of reads per sample was not reported which goes a long way to assuaging a knowledgeable reader's concerns about sequencing depth, especially when paired with the alpha diversity analysis also shown only in response to reviewers. The authors responded with the details that should be in the methods, but did not add them to the manuscript. Specifically to the point above, something like the following text:

"The sequencing run has generated in total 16x10^6 reads with the 77% of high quality reads (21% low quality, 2% test fragments). Finally we obtained 5.5x10^5 reads per sample (reads length were 244 bp mean, 260 bp median and 289 bp mode) and all analyzed specimens showed a suitable library’s profile. The analysis was performed by 16S Metagenomics GAIA 2.0 software and DESeq2 package software. Sequence data generated as FASTQ files, were analysed using the 16S Metagenomics GAIA 2.0 software which performs the quality control of the reads/pairs (i.e., trimming, clipping and adapter removal steps) through FastQC and BBDuk. The reads/pairs are mapped with BWA-MEM against the 16S databases (based on NCBI). Differential expression analysis using DESeq2 package to test for differential analysis by use of negative binomial generalized linear models was used. Only changes with FDR below 0.05 were considered significant" should be added to the methods, not just in the response to reviewers. Think of the reviewers as interested readers, if we both wanted this information, so will the readers. Similarly, the percent similarity used to determine species and genus calls, details on the reference sequence database used, etc. should be in the methods. In addition, it would be an improvement to include the CHAO1 alpha diversity plot in the response to reviewers in the supplemental figures. There is no word limit or figure limit in PLOS ONE, so there is no reason to leave out these details which provide the reader with key information to understand and trust the results.

Re: We apologize for missing to add this information within the text. We have now reported all the information contained in our previous response within the manuscript. The percent similarity used to determine species and genus calls was 93% at genus, 97% at species as reported in the new version of the text. Chao1 index is reported in figure 1.

In addition, it is still unclear how the analysis was performed on the reads, while the methods paragraph quoted above would suffice, later in the response to reviewers (reviewer 2, point 9) Ion Reporter is mentioned and strongly implies the reads were processed in some way on that platform. However, no mention of Ion Reporter is made in the manuscript. In addition, use of Ion Reporter would explain how the identity calls made from the multiple hypervariable regions were combined, but the authors still have not addressed this question in the response to reviewers or the manuscript (reviewer 1, point 1b).

RE: We apologize for the lack of clarity. Ion Reporter is one of the software that could be used for 16S analysis. However, Ion Reporter software was not mentioned in the manuscript because it is an analysis that has not been done for our sequences. For our whole study, we utilized the 16S Metagenomics GAIA 2.0 software and DESeq2 package software. Specifically, sequence data generated as FASTQ files, were analyzed using the 16S Metagenomics GAIA 2.0 software which performs the quality control of the reads/pairs (i.e., trimming, clipping and adapter removal steps) through FastQC and BBDuk. The reads/pairs are mapped with BWA-MEM against the 16S databases (based on NCBI). Differential expression analysis using DESeq2 package to test for differential analysis by use of negative binomial generalized linear models was used.

Ion Reporter was only mentioned in response to Reviewer 2 who asked us if it was possible to calculate the abundance coverage-based estimator (ACE) richness and evenness analyses. 

Lastly, In response to reviewer 1, point 3 the authors state that discussion of the strikingly absence of Bacteriodetes in one patient has been added to the text, however, this reviewer cannot find it in the revised manuscript and a search of the revised manuscript for the word "Bacteriodetes" returned no results. If the text has indeed been added, the authors should refer to specific line numbers.

RE: We thank the reviewer for this comment. We already reported (line 231-235) that growing evidence has shown that perturbation of the gut microbial community may fuel blooms of otherwise low abundance and harmful bacteria which can further exacerbate the intestinal inflammation. Indeed, dysbiosis in the distal gut is often characterized by a decrease in the prevalence of strict anaerobes and an increased relative abundance of facultative anaerobic bacteria.

For these reasons now we add thisese sentences in the revised manuscript (line 228-231):’Notably, a clear case of profound dysbiosis (Fig. 5) was observed in one ward patient (90-year-old patient with diabetes, meningioma and osteoporosis in association with an increase of C-Reactive Protein (CRP) and lymphocytes), with a significant increase in Proteobacteria and a relevant reduction in Bacteriodetes, reflecting an important inflammatory state’.

Reviewer #2: While the authors have addressed many of the initial concerns in the revised manuscript, there are still some issues that need to be further addressed (some of which were in the initial major concerns/comments previously):

1. The Supplemental Figures S1 and S2 need to be moved to the main body of the manuscript, instead of being placed in the Supplemental files. These are extremely important figures for transparency of the major limitations of this study (small sample size, antibiotic use, no pre- vs. post data, etc.). The reader should not have to seek this information in the Supplemental Files, since they pertain to the main findings of the study, especially since individual variation seemed to drive the significant response in some of the reported outcomes within study groups.

RE: We thank the reviewer for this comment and we have now moved Figure S1 and S2 (now renamed Figure 5 and Figure 6) within the manuscript as suggested. 

2. The authors should address in the Discussion (at a minimum) whether age, gender, co-morbidities, and/or antibiotic treatment (n=11 on study) altered individual 16S microbiota profiles reported.

RE: As suggested we have now added few sentences in the discussion section about the link between clinical data and the microbiota profiles. As for antibiotic treatment, we reported more details in lines 257-263. We recognize the potential effect of clinical/demographic data on the changes of mirobiota profile however we wish to underline that these data were not significantly different among the three groups.

---

## [Editor Report · Decision Letter 2]

1 Feb 2021

16S rRNA Gene Sequencing of Rectal Swab in Patients Affected by COVID-19

PONE-D-20-29220R2

Dear Dr. Di Caro,

We’re pleased to inform you that your manuscript has been judged scientifically suitable for publication and will be formally accepted for publication once it meets all outstanding technical requirements.

Kind regards,

Jane Foster, PhD

Academic Editor

PLOS ONE
---

## [Editor Report · Acceptance letter]

5 Feb 2021

PONE-D-20-29220R2 

16S rRNA Gene Sequencing of Rectal Swab in Patients Affected by COVID-19 

Dear Dr. Di Caro:

I'm pleased to inform you that your manuscript has been deemed suitable for publication in PLOS ONE. Congratulations! Your manuscript is now with our production department. 

Kind regards, 

on behalf of

Dr. Jane Foster 

Academic Editor

PLOS ONE